# InfoScale: Unleashing Training-free Variable-scaled Image Generation via Effective Utilization of Information

**Guohui Zhang**[1]*, **Jiangtong Tan**[1]*, **Linjiang Huang**[2], **Zhonghang Yuan**[1], **Mingde Yao**[3],
**Jie Huang**[1], **Feng Zhao**[1]†
[1] *MoE Key Lab of BIPC, University of Science and Technology of China* [2] *Beihang University* [3] *CUHK MMLab*

**Reviewed on OpenReview:** *https://openreview.net/forum?id=iB8PzxdpMf*

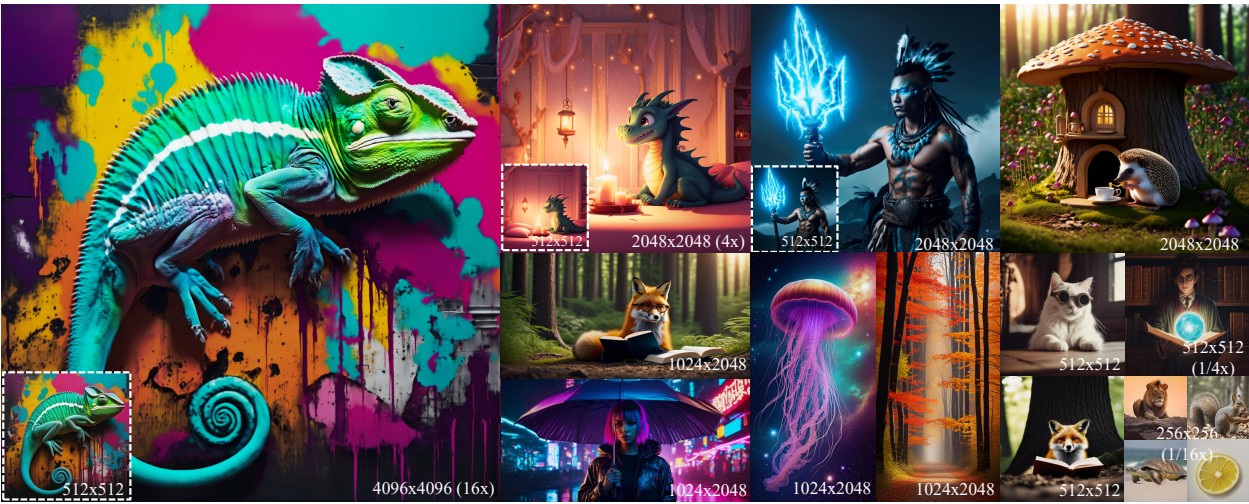

Figure 1: Generated images of InfoScale based on SDXL from lower resolution to higher resolution. Our method extends SDXL to generate images from $1/16\times, 1/4\times$ to $4\times, 16\times$ without any fine-tuning. Best viewed ZOOMED-IN.

## Abstract

Diffusion models (DMs) have become dominant in visual generation but suffer a performance drop when tested on resolutions that differ from the training scale, whether lower or higher. Current training-free methods for DMs have shown promising results, primarily focusing on higher-resolution generation. However, most methods lack a unified analytical perspective for variable-scale generation, leading to suboptimal results. In fact, the key challenge in generating variable-scale images lies in the differing amounts of information across resolutions, which requires information conversion procedures to be varied for generating variable-scaled images. In this paper, we investigate the issues of three critical aspects in DMs for a unified analysis in variable-scaled generation: dilated convolution, attention mechanisms, and initial noise. Specifically, 1) dilated convolution in DMs for the higher-resolution generation loses high-frequency information. 2) Attention for variable-scaled image generation struggles to adjust the information aggregation adaptively. 3) The spatial distribution of information in the initial noise is misaligned with the variable-scaled image. To solve the above problems, we propose **InfoScale**, an information-centric framework for variable-scaled image generation by effectively utilizing information from three aspects correspondingly. For information

---

*Equal contribution.

†Corresponding author

loss in 1), we introduce a Progressive Frequency Compensation module to compensate for high-frequency information lost by dilated convolution in higher-resolution generation. For information aggregation inflexibility in 2), we introduce an Adaptive Information Aggregation module to adaptively aggregate information in lower-resolution generation and achieve an effective balance between local and global information in higher-resolution generation. For information distribution misalignment in 3), we design a Noise Adaptation module to re-distribute information in initial noise for variable-scaled generation. Our method is plug-and-play, and extensive experiments demonstrate its effectiveness in variable-scaled image generation. Code is available at `https://github.com/zghhui/InfoScale`.

# 1 Introduction

Diffusion models (DMs) have witnessed remarkable progress in visual generation (Rombach et al., 2022; Ho et al., 2020; Song et al., 2020), which converts the information from initial noises to image space. Despite the powerful generation capabilities of DMs, for variable-scaled image generation that is often required in practical applications, directly inputting initial noise with resolutions lower or higher than the training resolution setting usually leads to visual defects, such as incomplete content in lower-resolution images and distorted structure in higher-resolution images. This is a challenging issue since the information conversion procedures are different in variable-scaled image generation, and the components in DMs (i.e., convolution, attentions) are over-optimized to process the information in training settings. Although fine-tuning DMs is a choice, it requires substantial computation resources and high-quality data.

Recently, quantities of training-free approaches for variable-scaled image generation have emerged and have attracted widespread attention. **Primarily**, most of them are focused on higher-resolution generation (He et al., 2023; Huang et al., 2024; Du et al., 2024; Qiu et al., 2024; Zhang et al., 2023). One line of training-free-based higher-resolution generation methods primarily relies on incorporating dilated convolution or samplings (i.e., ScaleCrafter (He et al., 2023) and FouriScale (Huang et al., 2024)) to align higher-resolution image structure information with the training resolution. While another line focuses on firstly generating the main structure in the training resolution and refining the details in the higher resolution (i.e., DemoFusion (Du et al., 2024) and FreeScale (Qiu et al., 2024) ), which also requires dilated convolution or samplings in higher-resolution generation procedures to avoid artifacts appearing. However, the dilated convolution used in these methods often leads to details information losses in higher-resolution generation. **Meanwhile**, Only a few works (Jin et al., 2023; Haji-Ali et al., 2024) focus on variable-scaled (**both lower and higher resolution**) image generation. However, their generated results often lack sufficient detail or require large latency overheads.

Although the aforementioned approaches have made great efforts, a unified analytical perspective for variable-scaled (both lower and higher resolution) generation has rarely been discussed. In fact, the key challenge in generating variable-scaled images is that the amount of information in the generated image is varied across resolutions, as shown in Fig. 2a. Higher-resolution images or latents generally contain a greater amount of information and larger proportions of high-frequency components, while lower-resolution ones behave oppositely. Since DMs are only optimized to convert the initial noise to the generated image at the level of training-resolution information amount, the contradiction in information conversion procedures across resolutions constrains the potential of applying DMs in generating variable-scaled images.

In this paper, we investigate the problems of three critical aspects in DMs for a unified analysis in variable-scaled generation: dilated convolution, attention mechanisms, and initial noise. Specifically, **1)** In higher-resolution generation, directly applying dilated convolution loses some high-frequency information, as shown in Fig. 2b, which prevents generating more image details. **2)** In lower-resolution generation, scaled attention struggles to aggregate enough information in the limited contextual range. However, in higher-resolution generation, attention tends to aggregate redundant and repetitive information, as shown in Fig. 3. This inflexible information aggregation leads to unreasonable information utilization. **3)** In lower-resolution generation, DMs struggle to handle non-uniform information distribution (i.e., lower entropy) in initial noise compared to the training setting, resulting in incomplete content. In higher-resolution generation, the infor-

mation distribution of initial noise is over-uniform and contains multiple responses to the prompt, causing the information in these regions to be processed independently into repetitive objects, see Fig. 4.

Therefore, in order to achieve effective utilization of information without the aforementioned three information utilization obstacles, we propose **InfoScale**, an information-centric variable-scaled image generation framework, including three key designs corresponding to them, respectively. Specifically, for the information loss in **1)**, we introduce a Progressive Frequency Compensation (PFC) module to extract high-frequency components from cached noise of the previous timestep to compensate for the predicted noise at the current timestep when applying dilated convolution in higher-resolution generation. For information aggregation inflexibility in **2)**, we introduce the Adaptive Information Aggregation (AIA) module to adaptively adjust the information aggregation ability of attention. We design dual-scaled attention (DSAttn) based on the original scaled attention by adjusting the attention entropy to be more adaptive, enhancing the information aggregation ability of attention in lower-resolution generation. We further fuse the features of DSAttn and original attention to effectively balance local-enhanced information (aggregated by DSAttn) and global information (original Attn) in higher-resolution generation. For information distribution misalignment in **3)**: we introduce the Noise Adaptation (NA) module, which enhances the uniformity of information distribution in the central region to encourage information aggregation in lower-resolution generation. We gradually suppress the uniformity of information distribution from centric to the surrounding region through the NA module to alleviate the repeated response to the prompt in higher-resolution generation. The designs of our method are training-free and are flexibly plug-and-play for DMs. Extensive experiments demonstrate that our framework significantly improves the visual quality in variable-scaled image generation.

Our core contributions can be summarized as follows:

- We propose **InfoScale**, an information-centric variable-scaled image generation framework, offering a unified analytical perspective for variable-scaled image generation.

- We design progressive frequency compensation, adaptive information aggregation, and noise adaptation modules to achieve efficient information utilization.

- Extensive experiments validate the effectiveness of our framework as a training-free plug-in for DMs.

## 2 Related Work

### 2.1 Text-to-image generation

Diffusion models (DMs) (Dhariwal & Nichol, 2021; Li et al., 2024; Liu et al., 2024a; Zhuo et al., 2024; Li et al.; Ganz & Elad, 2024) have attracted widespread attention due to their excellent image generation quality. Denoising Diffusion Probabilistic Models (DDPM) (Ho et al., 2020) demonstrated the potential of DMs in image generation. Moreover, Classifier-Free Guidance (CFG) (Ho & Salimans, 2022) ennobled DMs to generate images conforming to given prompts. Since operations in pixel space require substantial computational resources, Latent Diffusion Models (LDM) (Rombach et al., 2022) proposed to transfer the diffusion process to latent space (Blattmann et al., 2023; He et al., 2022), thereby reducing the training burden and laying the foundation for high-resolution image generation. Thanks to large-scale training data (Schuhmann et al., 2022), the Stable Diffusion series (Podell et al., 2023; Rombach et al., 2022) has achieved groundbreaking progress in visual generation.

### 2.2 Variable-scaled image generation

Due to being trained on limited resolutions, directly applying pre-trained diffusion models to generate images with novel resolutions often results in visual defects, such as incomplete content at lower-resolution images and repeated objects or distorted structures at higher-resolution images. For higher-resolution generation, some approaches propose training or fine-tuning models with higher-resolution images to improve the performance of models (Hoogeboom et al., 2023; Liu et al., 2024b; Ren et al., 2024; Chen et al., 2024). However, the scarcity of high-resolution image data and the significant increase in computational resource demands

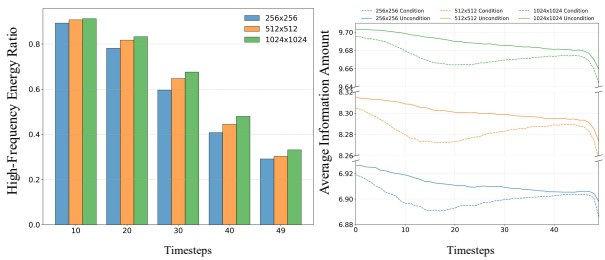 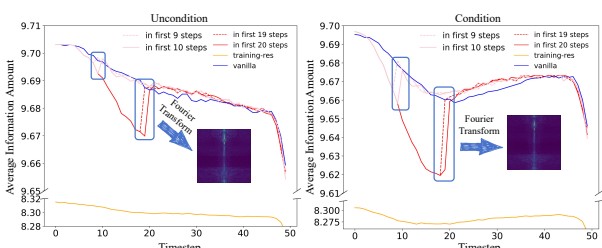

(a) Comparison of the ratio of high-frequency components and the information at different resolutions. Higher-resolution images contain a greater proportion of high-frequency components and larger information. Diffusion model trained at a fixed resolution can only correctly generate images containing the corresponding information, while issues occur with other resolutions.

(b) Information loss in dilated convolution. During the steps using dilated convolution, the information amount shows a significant decrease, indicating that dilated convolution reduces redundant information, while frequency analysis shows that this information includes some high-frequency components. Vanilla refers to no dilated convolution.

Figure 2: Analysis of image information and high-frequency components across different image resolutions.

due to resolution scaling limit the applicability of such methods. Many training-free approaches propose using specific strategies during inference to fully leverage the potential of diffusion models in higher-resolution image generation (Hwang et al., 2024; Lee et al., 2023; Kim et al., 2024; Lin et al., 2024a; Yang et al., 2024; Zhang et al., 2024; Lu et al., 2023; Yang et al., 2025). ScaleCrafter (He et al., 2023) and FouriScale (Huang et al., 2024) achieve structural consistency across different resolution by incorporating dilated convolution, while HiDiffusion (Zhang et al., 2023) dynamically resizes features to align with the training resolution. Nevertheless, these methods still suffer from degraded image details. MultiDiffusion (Bar-Tal et al., 2023) extends to larger resolutions by generating overlapping patches. DemoFusion (Du et al., 2024), AccDiffusion (Lin et al., 2024b), and FreeScale (Qiu et al., 2024) all first generate images at the training resolution to provide guidance for higher-resolution generation, yet their requires dilated convolution or samplings in higher-resolution generation procedures to avoid artifacts appearing. challenging. For variable-scaled image generation including lower-resolution generation, Attn-SF (Jin et al., 2023) adjusts attention entropy to achieve variable-scaled image generation, which has much space for improvement.

# 3 Method Motivation and Discussion

## 3.1 Information Loss

Information entropy (Shannon, 1948) is a fundamental concept in information theory. In this work, we calculate it based on the self-attention scores from DMs, which quantifies the information conversion procedures across different resolutions. The information entropy $H(X)$ is defined as:

$$H\left(\boldsymbol{X}\right) = -\sum_{i=1}^{n} p\left(x_i\right) \log p\left(x_i\right),$$ (1)

where $X$ is attention score after Softmax operation and $i$ represents each position.

Directly scaling pre-trained diffusion models to higher resolutions results in generating images with repetitive objects. This is because the model needs to process more information for higher-resolution generation, as shown in Fig. 2a. To mitigate this repetition issue, Scalecrafter (He et al., 2023) replaces the standard convolution in U-Net with dilated convolution with a large receptive field, which has proven to be effective and is commonly used by further higher-resolution generation works (i.e., FourierScale (Huang et al., 2024) and FreeScale (Qiu et al., 2024)), but it inevitably leads to image quality degradation with details losses. This raises the question of what causes this phenomenon.

We conduct experiments on 100 randomly generated prompts in higher-resolution generation to investigate the impact of dilated convolutions. We analyze the changes in information entropy during the sampling

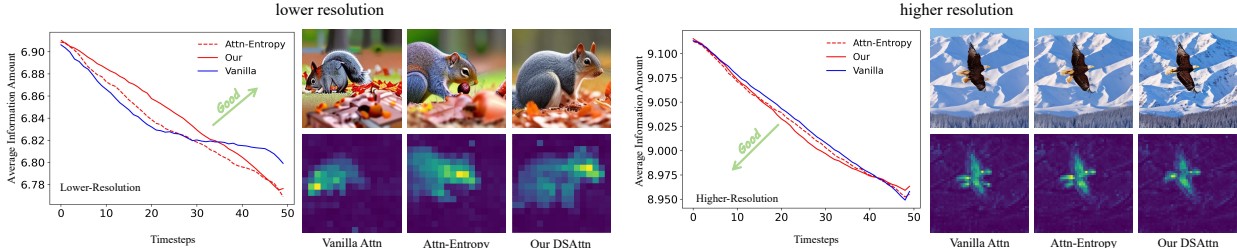

Figure 3: The top and bottom figures illustrate the inflexible aggregation ability of the model for lower- and higher-resolution generation, respectively. We use dual-scaled factors to achieve wider aggregation for lower-resolution generation, and vice versa for higher-resolution generation.

under different configurations. As shown in Fig. 2b, compared to standard convolution, dilated convolution tends to align the information to the training resolution in both conditional and unconditional sampling. This suggests that dilated convolution effectively compresses redundant information in higher-resolution generation, contributing to generating correct structures.

To further analyze the difference in information in predicted noise using dilated convolutions in the early stages, we performed frequency analysis for both unconditional and conditional predicted noise. The results show that, compared to standard convolutions, dilated convolutions cause the predicted noise to lose high-frequency information. In fact, high-frequency components generally occupy a larger proportion in high-resolution images compared to low-resolution images, as shown in Fig. 2a. Therefore, the loss of this high-frequency information potentially harms the quality of the generated image. To take advantage of dilated convolution in reducing redundant information while mitigating the loss of high-frequency information, frequency compensation is required to address this critical information-utilization bottleneck, see Sec. 4.1.

### 3.2 Information Aggregation Inflexibility

For DMs, the scaled dot-product attention allows the model to focus on prominent parts of the input during the generation, facilitating efficient information aggregation. Considering that this aggregation can be effectively reflected in the attention map, the high response area indicates that more visual and textual information is aggregated. We conduct experiments on information entropy and attention maps to analyze, see Fig. 3. The results indicate that the self-attention focused solely on narrower local regions at lower-resolution images, failing to fully utilize global information to generate complete content. Meanwhile, the self-attention attempts to attend to more redundant and repetitive information in higher resolutions, leading to the generation of repeated structures. In comparison, Attn-SF (Jin et al., 2023) applies adaptive aggregation of information according to resolutions, but it is still limited to only scaling the value inside the Softmax. This approach leaves room for further improvements in scaling self-attention, as it fails to fully exploit the potential of adaptive aggregation that could dynamically adjust across different resolutions (see Sec. 4.2).

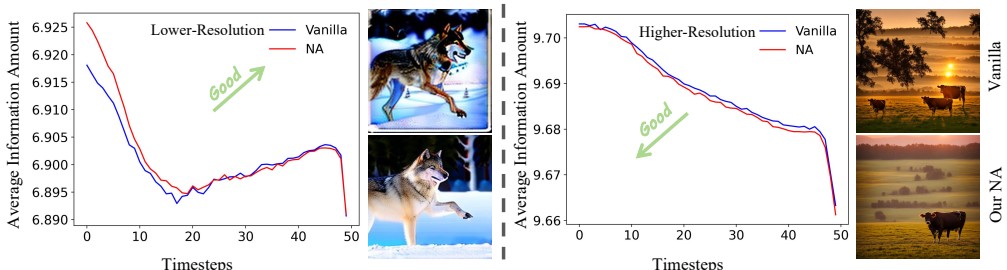

Figure 4: The left figure illustrates that increasing the variance to adjust the information distribution of the initial noise promotes the information aggregation in lower resolution generation. The high-resolution generation in the right figure is the opposite.

### 3.3 Information Distribution Misalignment

The distribution of initial noise is important for DMs. InitNo (Guo et al., 2024) reveals that initial noise includes both semantically consistent and inconsistent components. Inspired by this, we focus on the spatial distribution of information in initial noise. For higher resolution, the spatial information distribution of initial noise is more homogeneous than the training resolution, which contains multiple regions that respond to prompts, mainly manifested as multiple peaks in spatial distribution. For lower resolution, the spatial information distribution of initial noise is less homogeneous than the training resolution, which struggles to form a completely effective region to respond to the prompts, resulting in incomplete content. We conducted experiments in Fig. 4 that appropriately adjust the spatial distribution of the central area with variance scaling to align the training distribution. The experimental results show that this alignment improves information entropy, generating lower-resolution images with complete structures and higher-resolution images without repetitive objects. Therefore, we can further improve the utilization of information by adjusting a more appropriate initial noise distribution (see Sec. 4.3).

## 4 Method

Guided by the three analyses presented above, we propose **InfoScale**, a training-free framework comprising three dedicated modules. Specifically, **(1) Progressive Frequency Compensation** (PFC, Sec. 4.1) module recovers high-frequency components lost in dilated convolutions; **(2) Adaptive Information Aggregation** (AIA, Sec. 4.2) module employs a dual-scaled attention mechanism to adaptively control the receptive coverage of self-attention across resolutions; and the **(3) Noise Adaptation** (NA, Sec. 4.3) module modulates the initial noise distribution through a Gaussian-weighted mask to rectify spatial information imbalance.

### 4.1 Progressive Frequency Compensation

Consider the hidden feature $h$ and a convolution layer $f$ that it will pass through, where the convolution kernel is $k$. The dilated convolution operation $\Phi_d(\cdot)$ can be represented as:

$$f_k^d(h) = h \circledast \Phi_d(k), (h \circledast \Phi_d(k))(o) = \sum_{s+d \cdot t = p} h(p) \cdot k(q), \tag{2}$$

where $o, p, q$ are spatial locations used to index the feature and kernel, $\circledast$ denotes a convolution operation. This operation is equivalent to incorporating a down-sampling process before an up-sampling process, resulting in the loss of high-frequency information (Huang et al., 2024). Here, we aim to minimize the high-frequency information loss during the early stages of using dilated convolution.

We propose the Progressive Frequency Compensation module to address the loss of high-frequency information. Considering the cumulative frequency loss in the iteration process of the diffusion model and the continuity between consecutive latents, when predicting current noise $\epsilon_t$, we naturally employ the noise predicted in the previous step $\epsilon_{t-1}$ as the compensation for high-frequency information, which retains richer high-frequency information unaffected by dilated convolution. We identically process the conditional and unconditional noise. Denoting $N = \{unc, c\}$, it can be shown as:

$$\epsilon_{t-1}^{N,h} = \mathcal{FFT}(\epsilon_{t-1}^N) \odot (1 - \mathcal{H}), \quad \epsilon_t^{N,l} = \mathcal{FFT}\left(\epsilon_t^N\right) \odot \mathcal{H}, \quad \tilde{\epsilon}_t^N = \mathcal{IFFT}(\epsilon_{t-1}^{N,h} + \epsilon_t^{N,l}), \tag{3}$$

where $\mathcal{FFT}$ is the Fast Fourier Transform and $\mathcal{IFFT}$ is the Inverse Fast Fourier Transform. $\mathcal{H}$ is the low-pass filter (LPF) with stop frequency of $D_0 = 0.25$ by default. More details can be found in the appendix.

### 4.2 Adaptive Information Aggregation

In order to adaptively adjust the aggregation degree of information in self-attention, we propose Dual-Scaled Attention (DSAttn). Specifically, following prior work (Jin et al., 2023), we introduce a scaled factor $\alpha$ within the Softmax operation in the self-attention layer, determined by the number of tokens in the current attention layer during both training and inference phases. Additionally, for low-resolution image generation,

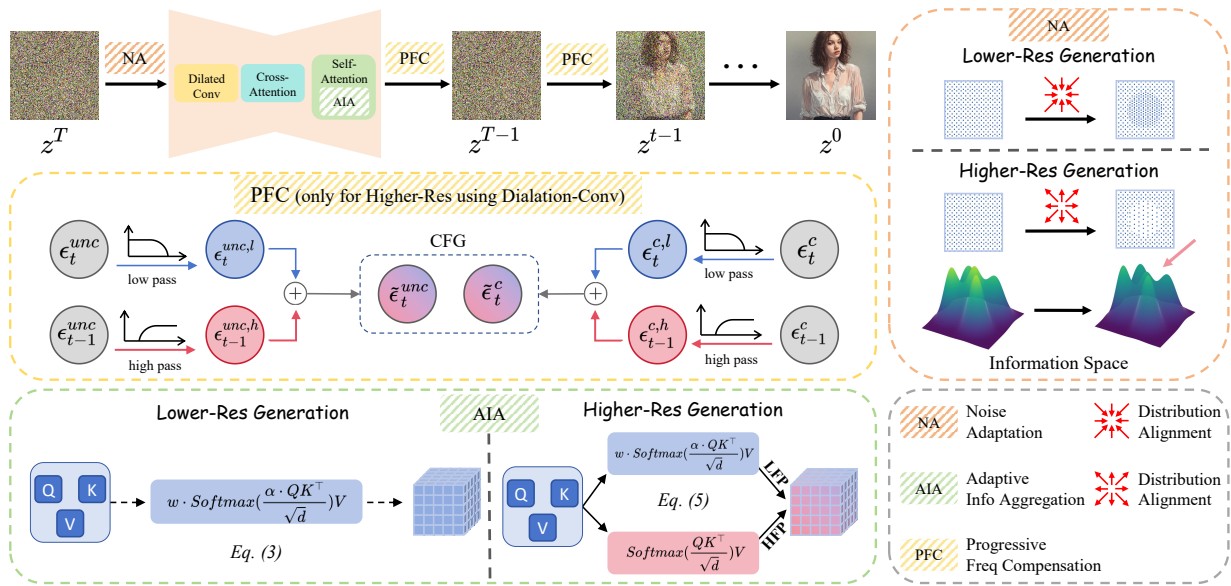

Figure 5: Overall framework of InfoScale. (a) In higher-res (resolution) generation, the Noise Adaptation (NA) module first modulates the initial noise according to resolution. Then, the Progressive Frequency Compensation (PFC) module extracts high-frequency components from cached noise of the previous timestep to compensate for the predicted noise at the current timestep when applying dilated convolution. The Adaptive Information Aggregation module further fuses local (blue) and global information (red). (b) In lower-res generation, we also usethe NA module and replace the original self-attention layer with DSAttn.

we incorporate an empirical hyperparameter $c_w$ to rescale the entire attention feature. This stems from our empirical observation that dual-scaled factors allow attention to cover a broader range, thereby achieving better visual generation quality, which is shown below:

$$\text{DSAttn(Q, K, V)} = w \cdot \text{Softmax}(\frac{\alpha \cdot QK^\top}{\sqrt{d}})V, \tag{4}$$

where $a$ and $w$ are scaled factors, $d$ is token dimension. Specifically, we calculate $\alpha$ and $w$ in following form:

$$\alpha = \sqrt{\log_L N}, \quad w = \begin{cases} c_w & \text{L} > \text{N} \\ 1 & \text{L} \le \text{N} \end{cases}, \tag{5}$$

where $L, N$ are the number of tokens in self-attention in the training and testing phase. $c_w$ is a hyperparameter with the value 0.75. In practice, we adopt different strategies for low- and high-resolution generation.

**Lower-Resolution Generation.** We only replace the original attention layer with DSAttn in lower resolution generation to improve the ability to aggregate information.

**Higher Resolution Generation.** After using PFC module and DSAttn, the details of local features are effectively enhanced. To further leverage the original attention advantage in processing global information, we further fuse features from DSAttn and original attention to enhance global details and local structure. Specifically, we extract the low-frequency and high-frequency components of the features computed by DSAttn and the original attention layer, respectively. The low-frequency component is obtained by downsampling and then upsampling the feature, while the high-frequency component is obtained by subtracting the transformed result. Then, we fuse these two parts to obtain the enhanced feature.

$$\tilde{h}_t = \mathcal{U}\left(\mathcal{D}(h_t^s)\right) + \left(h_t - \mathcal{U}\left(\mathcal{D}(h_t)\right)\right), \tag{6}$$

where $\mathcal{U}$ and $\mathcal{D}$ represent upsampling and downsampling operations (Nearest Neighbor interpolation).

Table 1: Quantitative results. - indicates no data available for this metric. +Our is plug-and-play

| Method | Factor | SD1.5 | | | | SD2.1 | | | | SDXL | | | | | |
|---|---|---|---|---|---|---|---|---|---|---|---|---|---|---|---|
| | | FID↓ | KID↓ | FID$_c$↓ | KID$_c$↓ | FID↓ | KID↓ | FID$_c$↓ | KID$_c$↓ | FID↓ | KID↓ | FID$_c$↓ | KID$_c$↓ | Clip↑ | Time↓ |
| Direct-Inference | 0.5×0.5 | 114.76 | 0.031 | - | - | 101.79 | 0.026 | - | - | 96.31 | 0.017 | 78.72 | 0.021 | 30.26 | 3s |
| Attn-SF (Jin et al., 2023) | | 95.09 | 0.021 | - | - | 88.34 | 0.019 | - | - | 71.85 | 0.014 | 49.20 | 0.009 | 30.62 | 3s |
| ElasticDiffusion (Haji-Ali et al., 2024) | | 94.40 | 0.030 | - | - | 92.16 | 0.028 | - | - | 70.45 | 0.011 | 49.96 | 0.009 | 30.98 | 50s |
| Ours | | 90.34 | 0.017 | - | - | 82.93 | 0.016 | - | - | 71.02 | 0.013 | 49.05 | 0.008 | 31.04 | 3s |
| Direct-Inf | 2×2 | 86.97 | 0.014 | 46.45 | 0.009 | 80.82 | 0.011 | 40.56 | 0.007 | 117.07 | 0.033 | 128.05 | 0.041 | 31.45 | 35s |
| Attn-SF (Jin et al., 2023) | | 82.28 | 0.011 | 45.45 | 0.007 | 79.81 | 0.010 | 37.87 | 0.006 | 111.86 | 0.030 | 124.17 | 0.035 | 31.55 | 35s |
| HiDiffusion (Zhang et al., 2023) | | 75.00 | 0.009 | 44.20 | 0.008 | 66.96 | 0.006 | 38.13 | 0.007 | 104.62 | 0.024 | 108.32 | 0.025 | 31.92 | 19s |
| MegaFusion (Wu et al., 2025) | | 67.43 | 0.008 | 38.92 | 0.007 | 64.11 | 0.005 | 37.09 | 0.007 | 72.38 | 0.007 | 93.06 | 0.018 | 32.47 | 18s |
| DiffuseHigh (Kim et al., 2024) | | - | - | - | - | - | - | - | - | 60.87 | 0.004 | 84.33 | 0.015 | 32.96 | 40s |
| DemoFusion (Du et al., 2024) | | - | - | - | - | - | - | - | - | 54.25 | 0.003 | 71.69 | 0.013 | 33.58 | 90s |
| Accdiffusion (Lin et al., 2024b) | | - | - | - | - | - | - | - | - | 55.34 | 0.003 | 76.15 | 0.008 | 33.69 | 98s |
| FreCaS (Zhang et al., 2024) | | - | - | - | - | - | - | - | - | 54.01 | 0.003 | 62.50 | 0.007 | 33.99 | 23s |
| FouriScale (Huang et al., 2024) | | 68.81 | 0.008 | 39.79 | 0.007 | 65.22 | 0.006 | 38.19 | 0.007 | 78.17 | 0.017 | 93.75 | 0.025 | 32.22 | 65s |
| FouriScale (Huang et al., 2024) +Our | | 68.16 | 0.008 | 39.03 | 0.007 | 63.74 | 0.005 | 36.84 | 0.006 | 77.47 | 0.017 | 93.18 | 0.024 | 32.45 | 67s |
| ScaleCrafter (He et al., 2023) | | 69.02 | 0.008 | 40.72 | 0.007 | 64.93 | 0.006 | 37.70 | 0.006 | 73.14 | 0.012 | 91.26 | 0.021 | 32.98 | 38s |
| ScaleCrafter (He et al., 2023) +Our | | 66.34 | 0.007 | 37.97 | 0.006 | 62.73 | 0.004 | 36.47 | 0.006 | 72.57 | 0.012 | 91.03 | 0.021 | 33.08 | 40s |
| FreeScale (Qiu et al., 2024) | | - | - | - | - | - | - | - | - | 51.99 | 0.003 | 60.99 | 0.006 | 34.23 | 47s |
| FreeScale (Qiu et al., 2024) +Our | | - | - | - | - | - | - | - | - | 50.79 | 0.002 | 59.50 | 0.004 | 34.26 | 48s |
| Direct-Inf | 4×4 | 180.47 | 0.062 | 58.56 | 0.020 | 173.75 | 0.058 | 53.05 | 0.011 | 189.08 | 0.078 | 165.43 | 0.059 | 30.17 | 504s |
| Attn-SF (Jin et al., 2023) | | 169.05 | 0.054 | 57.72 | 0.018 | 174.72 | 0.059 | 51.90 | 0.009 | 187.24 | 0.079 | 161.68 | 0.056 | 30.79 | 504s |
| HiDiffusion (Zhang et al., 2023) | | 135.00 | 0.043 | 62.66 | 0.027 | 119.76 | 0.033 | 76.03 | 0.026 | 144.08 | 0.056 | 186.45 | 0.079 | 31.34 | 138s |
| DiffuseHigh (Kim et al., 2024) | | - | - | - | - | - | - | - | - | 75.43 | 0.022 | 115.40 | 0.031 | 32.07 | 557s |
| DemoFusion (Du et al., 2024) | | - | - | - | - | - | - | - | - | 60.60 | 0.006 | 94.81 | 0.019 | 32.46 | 861s |
| Accdiffusion (Lin et al., 2024b) | | - | - | - | - | - | - | - | - | 70.34 | 0.018 | 109.15 | 0.028 | 32.18 | 896s |
| FreCaS (Zhang et al., 2024) | | - | - | - | - | - | - | - | - | 65.19 | 0.015 | 94.55 | 0.019 | 32.21 | 130s |
| FouriScale (Huang et al., 2024) | | 76.63 | 0.011 | 57.19 | 0.019 | 75.09 | 0.009 | 55.48 | 0.019 | 113.25 | 0.033 | 161.24 | 0.062 | 31.64 | 654s |
| FouriScale (Huang et al., 2024) +Our | | 76.15 | 0.011 | 57.11 | 0.018 | 75.02 | 0.009 | 54.90 | 0.018 | 113.06 | 0.033 | 158.32 | 0.060 | 31.68 | 657s |
| ScaleCrafter (He et al., 2023) | | 70.46 | 0.008 | 53.41 | 0.016 | 76.11 | 0.011 | 55.99 | 0.020 | 119.86 | 0.036 | 172.25 | 0.070 | 31.40 | 693s |
| ScaleCrafter (He et al., 2023) +Our | | 70.37 | 0.007 | 52.52 | 0.015 | 75.47 | 0.010 | 55.69 | 0.020 | 119.10 | 0.035 | 170.98 | 0.068 | 31.47 | 698s |
| FreeScale (Qiu et al., 2024) | | - | - | - | - | - | - | - | - | 60.16 | 0.008 | 94.20 | 0.019 | 32.75 | 532s |
| FreeScale (Qiu et al., 2024)+Our | | - | - | - | - | - | - | - | - | 59.74 | 0.007 | 92.82 | 0.017 | 32.88 | 534s |

## 4.3 Noise Adaptation

We design a noise adaptation module to align the information distribution in the initial noise without incurring additional computational burden. Specifically, we modulate the initial noise $Z_T$ to obtain adaptive noise $\hat{Z}_T$ using a mask $W$ with Gaussian weights. For different resolutions, W has different weights.

$$\hat{Z}_T = W \odot Z_T. \tag{7}$$

**Lower-Resolution Generation.** The weight of $W$ increases from the center to the surrounding to concentrate the information in the central area, which aims to generate a complete object.

**Higher Resolution Generation.** The weight of $W$ decreases from the center to the surroundings, which aims to mitigate distorted structure and repetitive objects.

## 5 Experiments

**Experimental Settings.** To demonstrate the effectiveness of our method, we perform evaluation on SD1.5 (Rombach et al., 2022), SD2.1 (Rombach et al., 2022) and SDXL (Podell et al., 2023). We perform three unseen resolutions, with scaling factors of $0.25 \times 0.25$, $2 \times 2$, $4 \times 4$ relative to the original training resolution. Specifically, we generate images of $256 \times 256$, $1024 \times 1024$, and $2048 \times 2048$ for SD1.5 and SD2.1, while $512 \times 512$, $2048 \times 2048$ and $4096 \times 4096$ for SDXL. We randomly select 1024 prompts to conduct evaluation from LAION-5B (Schuhmann et al., 2022), which contains 5 billion image-caption pairs.

**Evaluation metrics.** Following prior work, we report Frechet Inception Distance (FID) (Heusel et al., 2017) and Kernel Inception Distance (KID) (Bińkowski et al., 2018) to evaluate the quality and diversity of generated images. Following previous work (Chai et al., 2022; Qiu et al., 2024), we use crop local patches to calculate the above metrics, defined as $FID_c$ and $KID_c$. Notably, for lower resolution generation on SD1.5 and SD2.1, the images cannot be further cropped, so they do not have $FID_c$ and $KID_c$. Additionally, we also report the CLIP score (Clip) (Radford et al., 2021) and inference time (Time).

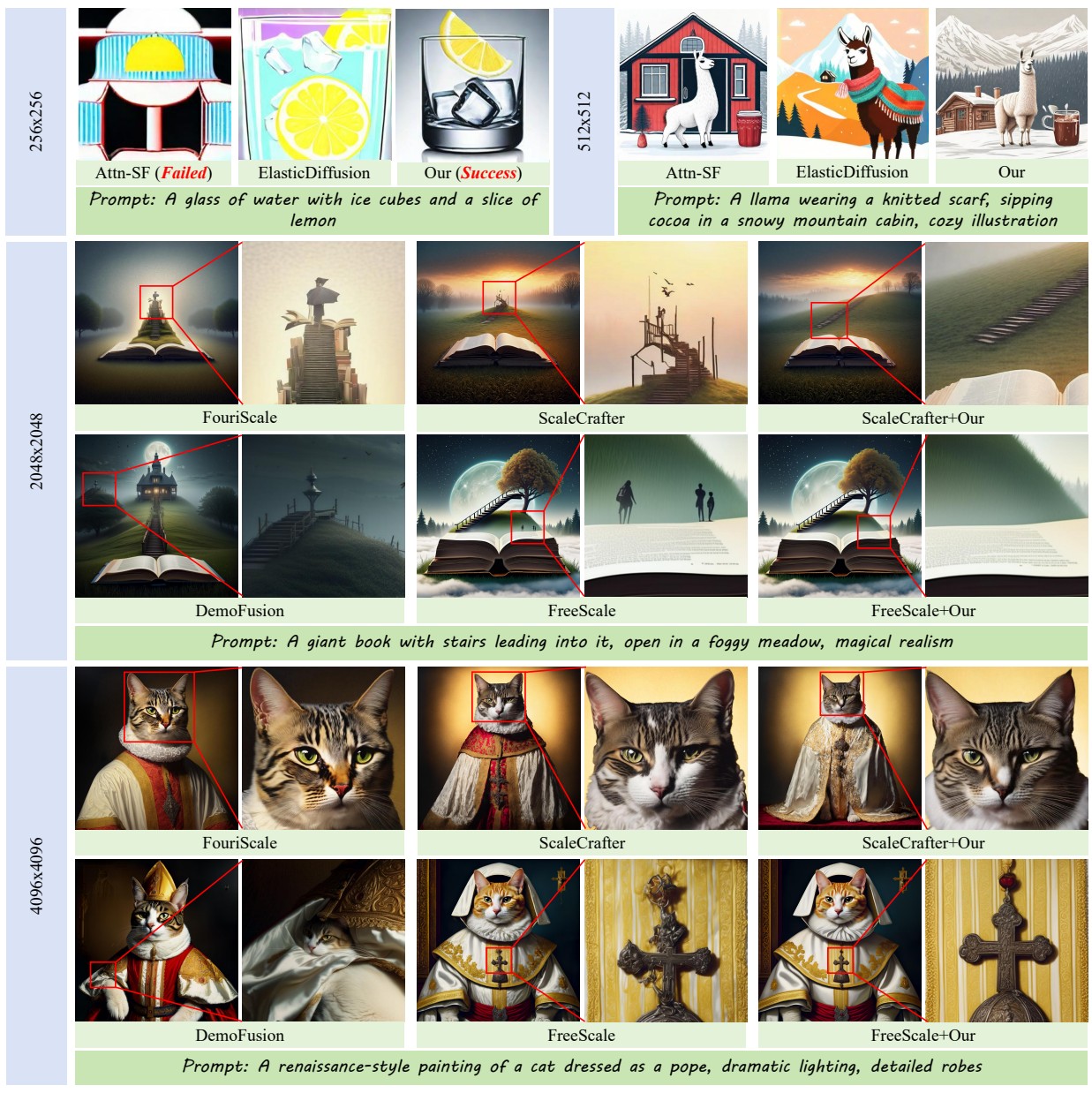

Figure 6: Qualitative comparison of images with other baselines. The training resolution of SDXL is $1024^2 (1/16\times)$. Our method generates complete visual content on lower resolution, even up to $256^2 (1/16\times)$. Best viewed ZOOMED-IN.

## 5.1 Main Results.

For lower resolution generation, we compare our method with SDXL (Podell et al., 2023) Direct-Inference, Attn-SF (Jin et al., 2023), and ElasticDiffusion (Haji-Ali et al., 2024). For higher resolution, we compare with SDXL (Podell et al., 2023) Direct-Inference, Attn-SF (Jin et al., 2023), ScaleCrafter (He et al., 2023), FouriScale (Huang et al., 2024), HiDiffusion (Zhang et al., 2023), AccDiffusion (Lin et al., 2024b), MegaFusion (without experimental setting at $4096^2$ resolutions) (Wu et al., 2025), DiffuseHigh (Kim et al., 2024), FreCaS (Zhang et al., 2024), DemoFusion (Du et al., 2024) and FreeScale (Qiu et al., 2024). Additionally, we integrate our method into (He et al., 2023; Huang et al., 2024; Qiu et al., 2024). **More results** on **higher-resolution**, and comparisons with **Super-Resolution** are available in the supplementary materials.

**Quantitative results.** In Table 1, for lower resolution generation, our method achieves the best performance in almost all metrics, demonstrating effectiveness in generating complete and detailed visual content. It's important to note that the inference times for ElasticDiffusion are approximately **15 times longer than** ours. For higher resolution generation, our method further enhances the performance of three baselines, particularly in terms of $FID_c$ and $KID_c$. In the $2 \times 2$ experiments on SD1.5 and SD2.1, ScaleCrafter+Our (integrated with our method) outperforms other approaches. In all experiments on SDXL, FreeScale+Our achieves the best scores on almost all metrics.

**Qualitative results.** In the Fig. 6, we show the visual comparison results. For lower resolution, the generated results of our method have richer details and a more complete structure compared with Attn-SF and ElasticDiffusion, including a smaller scaling factor ($0.25 \times 0.25$), which demonstrates the powerful ability of our method. For higher-resolution generation, our method further reduces the small local repetition that appears in Scalecrafter and freescale on the 2x2 experiment. For the 4x4 experiment, the cat's eyes and ears in ScaleCrafter have obvious aliasing, and FreeScale fails to generate an accessory on the chest. In comparison, our results have better visual quality.

Table 2: Ablation studies for each component in InfoScale

| PFC | AIA | NA | FID | KID | $FID_c$ | $KID_c$ |
|---|---|---|---|---|---|---|
| ✓ | | | 63.99 | 0.005 | 37.40 | 0.006 |
| ✓ | | ✓ | 63.48 | 0.005 | 37.08 | 0.007 |
| ✓ | ✓ | | 62.95 | 0.005 | 36.73 | 0.006 |
| ✓ | ✓ | ✓ | 62.73 | 0.004 | 36.47 | 0.006 |

## 5.2 Ablation Study

In this section, we use the SD2.1 and conduct a series of ablation experiments with 2x2 scale factor setting to verify the effectiveness of each component, as shown in Table 8.

**Effect of Progressive Frequency Compensation (PFC).** As shown in Fig. 7 (c). Although the dilated convolution significantly reduces the repetition issue, the background of the generated images becomes blurred. Compared with (a), our PFC compensatesfor the high-frequency information loss caused by the dilated convolution, making the background clearer and improving the global details.

**Effect of Adaptive Information Aggregation (AIA).** DSAttn has better information aggregation ability as shown in Fig. 3. We further utilize the AIA module in higher resolution to balance local and global information, as shown in Fig. 7 (d). After adopting AIA, the image details are further improved. The comparison results of DSAttn and AIA can be found in the supplementary material.

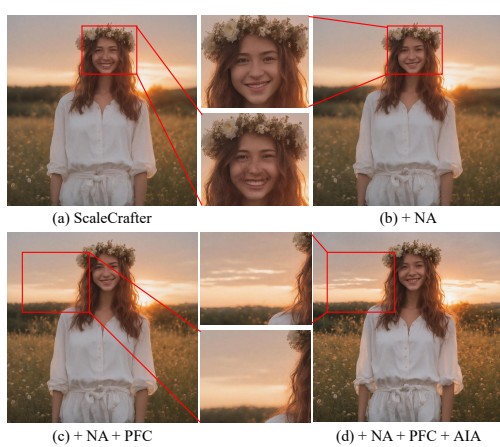

(a) ScaleCrafter  (b) + NA

(c) + NA + PFC  (d) + NA + PFC + AIA

Figure 7: Qualitative results for ablation study.

**Effect of Noise Adaptation (NA).** We use NA module to suppress the distribution of information in the central region to mitigate the phenomenon of repetitive content in higher resolution. As shown in Fig. 7 (b), the messy hair on the face is successfully removed after using the NA module.

## 5.3 The experiments on DiT

SD3 (Esser et al., 2024) is based on the DIT (Peebles & Xie, 2023) structure, which is different from Unet-based SD1.5 (Rombach et al., 2022) and SDXL (Podell et al., 2023). We performed experiments on SD3 $512 \times 512$, $2048 \times 2048$ and $3072 \times 3072$ settings to verify the generality of our method. As shown in the

Table 3: Quantitative Results on the SD3

| Method | Scale Factor | FID | KID | $\text{FID}_c$ | $\text{KID}_c$ |
|---|---|---|---|---|---|
| Direct-Inf | $0.5 \times 0.5$ | 85.49 | 0.028 | 79.06 | 0.020 |
| Our | | 84.16 | 0.025 | 78.34 | 0.018 |
| Direct-Inf | $2 \times 2$ | 71.24 | 0.017 | 67.56 | 0.025 |
| Our | | 70.38 | 0.016 | 67.12 | 0.023 |
| Direct-Inf | $3 \times 3$ | 129.58 | 0.041 | 117.24 | 0.035 |
| Our | | 128.70 | 0.039 | 116.42 | 0.034 |

Fig. 10 and Tab. 3, we observe that our method further improves the quality of image generation, achieving lower FID and KID scores.

## 6 Conclusion

We propose **InfoScale**, an information-centric variable-scaled image generation framework, achieving effective information utilization for DMs. We believe that information amount of the generated image is different across resolutions, leading to the information conversion procedures needing to be varied when converting the initial noise to variable-scaled images. We design the Progressive Frequency Compensation module, the Adaptive Information Aggregation module, and the Noise Adaptation module to address these challenges. Our method is plug-and-play for DMs, and extensive experiments demonstrate the effectiveness of our method.

## 7 Acknowledgments

This work was supported by the Anhui Provincial Natural Science Foundation under Grant 2108085UD12. We acknowledge the support of GPU cluster built by MCC Lab of Information Science and Technology Institution, USTC. The AI-driven experiments, simulations and model training were performed on the robotic AI-Scientist platform of Chinese Academy of Sciences.

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

# A Appendix

## A.1 Motivation for Variable-scaled Generation

Investigating variable-scaled generation is a significant research. Just as Attention-SF (Jin et al., 2023) and ElasticDiffusion (Haji-Ali et al., 2024) did lower- and higher-resolution generation, we follow their task setting. Additionally, (1) we need to discuss both lower- and higher-resolution generation to comprehensively explain the essence of information utilization in Diffusion models. (2) In Table 4, our method requires less inference time and GPU memory, especially on portable devices, with a partial trade-off in fidelity.

Additionally, we provide more low-resolution generated visual results on three different SD models: SD 1.5, SD 2.1, and SDXL, as shown in Fig. 11.

Table 4: Quantitative Results on the SDXL 512x512 setting

| Method | FID | Time | GPU Memory (GB) |
|---|---|---|---|
| Direct-Inf | 96.31 | 4s | 7.6 |
| SDXL+DownSampling | 58.44 | 9s | 10.5 |
| Our | 71.02 | 4s | 7.6 |

## A.2 Implementation Details

### A.2.1 Information Entropy

Information entropy (Shannon, 1948) is a fundamental concept in information theory. It is used to quantify the uncertainty or unpredictability of random variables or information. For a discrete random variable $X$, its entropy $H(X)$ is defined as:

$$H\left(\boldsymbol{X}\right) = -\sum_{i=1}^{n} p\left(x_i\right) \log p\left(x_i\right) \tag{8}$$

where $x_i$ is a possible outcome of the random variable $X$ and $p(x_i)$ indicates the probability of the $x_i$. Information entropy measures the average uncertainty of random variables. Due to the calculation characteristics of softmax, the attention score can be well used as the probability value in the information entropy formula. Each attention map can be regarded as a random variable. Therefore, information entropy can be linked to the attention mechanism (Li et al., 2025), where a larger entropy indicates that a wider range of contextual information is considered, while a smaller entropy otherwise. Based on the above analysis, we use the attention scores from the self-attention layer in the second-to-last block of the model to dynamically measure the information amount of the latent.

### A.2.2 High-Frequency Energy Ratio

We quantify the amount of high-frequency information in images of different resolutions by calculating the proportion of high-frequency component to the total energy.

### A.2.3 Noise Adaptation

In Noise Adaptation module, our $W$ is a Gaussian weight obtained from a 2D Gaussian function. Specifically, $\mu_x = \frac{h}{2}$, $\mu_y = \frac{w}{2}$, $\sigma_x = \frac{h}{K}$, and $\sigma_y = \frac{w}{K}$, where $h$ represents Height, $w$ represents Width, and $K$ represents KERNEL DIVISION, with a default value of 3.

### A.2.4 Experiments Setting

We perform DDIM (Song et al., 2020) sampling with 50 steps for all experiments, with the classifier-free guidance set to 7.5 by default. For the integration of FouriScale (Huang et al., 2024) and FreeScale (Qiu et al., 2024), we only use the Progressive Frequency Compensation module and Noise Adaptation module.

Specifically, for Freescale, all our modules operate at a higher resolution, with no processing applied to the first stage. Additionally, we note that Freescale performs interpolation operations in Self-Cascade Upscaling, and our experiments found that this interpolation affects the supplementation of high frequencies. Therefore, we made appropriate adjustments: during the time steps of dilated convolutions, we changed the interpolation operation to frequency fusion, similar to (Yang et al., 2024). The difference is that we used a fixed stop frequency, and we execute Freescale's default operations at timestamp without dilated convolution.

### A.2.5 Plug-and-Play

Our proposed method is plug-and-play for DMs and can be integrated into FreeScale (Qiu et al., 2024) to achieve effective utilization of information. Fig. 8 illustrates our integration on FreeScale (Qiu et al., 2024).

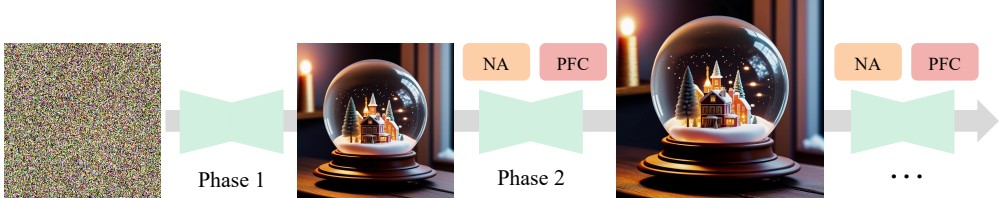

Figure 8: Plug-and-play of applying our method for FreeScale.

### A.2.6 Progressive Frequency Compensation

Regarding the loss of high-frequency information in dilated convolutions, the predicted noise at the current time step without dilated convolution is an ideal source of high-frequency components, but it is difficult to obtain unless noise prediction is performed again. We consider that the latent at adjacent timesteps have continuity, and the loss of high-frequency components in the noise from the previous step is relatively smaller compared to the noise at the current step. Therefore, we use the noise predicted at the previous time step as a compensation source of high-frequency information. In addition, we perform Adaptive Instance Normalization (AdaIN) before frequency fusion to align the statistics of the cached noise with the predicted noise at the current time step. This is due to the difference in the signal-to-noise ratio (SNR) (Hoogeboom et al., 2023) between them, and directly using the predicted noise from the previous step for frequency fusion can easily disrupt the distribution of the current predicted noise, causing out-of-distribution issues.

### A.3 More Results

### A.3.1 Ablation studies

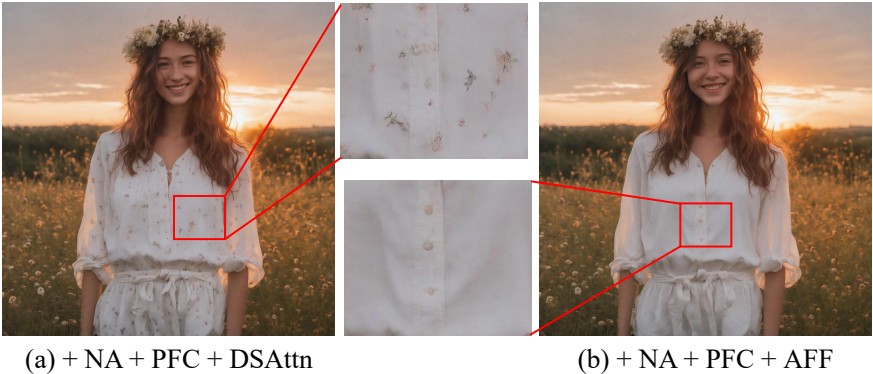

(a) + NA + PFC + DSAttn        (b) + NA + PFC + AFF

Figure 9: Comparison of DSAttn and AFF.

We experimentally find that directly utilizing Dual-Scaled Attention (DSAttn) in higher-resolution generation will cause some local content to appear in the generated image, as shown in the Fig. 9. As for the reason for this phenomenon, we believe that DSAtn weakens its ability to pay attention to global information because it focuses on local information. To this end, we perform Attention Feature Fusion(AFF) in higher-resolution generation. In contrast, by combining information at different scales, we achieved better visual effects.

### A.3.2  Experiments on other aspect ratio

We compare with SDXL (Podell et al., 2023) Direct-Inference, Attn-SF (Jin et al., 2023), ScaleCrafter (He et al., 2023), FouriScale (Huang et al., 2024), AccDiffusion (Lin et al., 2024b) and DemoFusion (Du et al., 2024) to verify the effectiveness of our method on other aspect ratios(2:4) , as shown in Table 5.

Table 5: Quantitative Results on the SDXL $2048 \times 4096$ setting

| Method | Scale Factor | FID | KID | $FID_c$ | $KID_c$ |
|---|---|---|---|---|---|
| Direct-Inf (Huang et al., 2024) | | 154.36 | 0.047 | 145.06 | 0.054 |
| Attn-SF (Jin et al., 2023) | | 150.63 | 0.045 | 139.85 | 0.050 |
| DemoFusion (Du et al., 2024) | | 57.44 | 0.011 | 81.86 | 0.024 |
| AccDiffusion (Du et al., 2024) | $2 \times 4$ | 56.87 | 0.010 | 81.19 | 0.013 |
| ScaleCrafter (He et al., 2023) | | 92.63 | 0.028 | 109.73 | 0.025 |
| ScaleCrafter (He et al., 2023)+Our | | 92.11 | 0.024 | 109.55 | 0.023 |
| FouriScale (Huang et al., 2024) | | 90.28 | 0.022 | 106.37 | 0.021 |
| FouriScale (Huang et al., 2024)+Our | | 89.79 | 0.020 | 106.02 | 0.020 |

### A.3.3  Comparison with Super-Resolution methods

We compare with super-resolution methods, including StableSR (Wang et al., 2024), ESRGAN (Wang et al., 2018) and Real-ESRGAN (Wang et al., 2021) on SDXL $512 \times 512$ and $2048 \times 2048$ settings to demonstrate effectiveness of our method, as shown in Table 6.

Table 6: Quantitative Results on the SDXL

| Method | Scale Factor | FID | KID | $FID_c$ | $KID_c$ |
|---|---|---|---|---|---|
| SDXL+StableSR (Wang et al., 2024) | | 58.53 | 0.003 | 64.29 | 0.008 |
| SDXL+Real-ESRGAN (Wang et al., 2021) | $2 \times 2$ | 60.32 | 0.004 | 65.67 | 0.012 |
| FreeScale (Qiu et al., 2024) | | 51.99 | 0.003 | 60.99 | 0.006 |
| FreeScale (Qiu et al., 2024) +our | | 50.79 | 0.002 | 59.50 | 0.004 |
| SDXL+ESRGAN (Wang et al., 2018) | | 61.85 | 0.012 | 98.08 | 0.024 |
| SDXL+Real-ESRGAN (Wang et al., 2021) | | 60.95 | 0.010 | 97.26 | 0.022 |
| SDXL+StableSR (Wang et al., 2024) | $4 \times 4$ | 59.93 | 0.008 | 95.97 | 0.021 |
| FreeScale (Qiu et al., 2024) | | 60.16 | 0.008 | 94.20 | 0.019 |
| FreeScale (Qiu et al., 2024) +our | | 59.74 | 0.007 | 92.82 | 0.017 |

### A.3.4  More results on GenEval

We compare with ScaleCrafter (He et al., 2023) on GenEval Benchmark (Ghosh et al., 2023) to demonstrate the effectiveness of our method, as shown in Table 7.

Table 7: Quantitative Results on the SDXL $2048 \times 4096$ setting

| Method | Resolution | Single Obj. | Two Obj. | Counting | Colors | Position | Color Attr. | Overall |
|---|---|---|---|---|---|---|---|---|
| ScaleCrafter (He et al., 2023) | 2048 | 0.89 | 0.56 | 0.25 | 0.72 | 0.09 | 0.18 | 0.45 |
| ScaleCrafter (He et al., 2023) +our | 2048 | 0.90 | 0.54 | 0.28 | 0.80 | 0.10 | 0.23 | 0.48 |

### A.3.5 The ablation experiments on stop-frequency in PFC

As shown in Table 8, we conduct an ablation study on the cutoff frequency of the PFC module. This module is designed to compensate for the loss of high-frequency details caused by dilated convolutions. The results reveal a critical trade-off: setting the cutoff frequency too high leads to insufficient high-frequency compensation. Conversely, setting it too low introduces excessive low-frequency content from the previous timestep, which disrupts the denoising process for the current step.

Table 8: Ablation studies for stop-frequency in PFC

| PFC | FID | KID | $\text{FID}_c$ | $\text{KID}_c$ |
|---|---|---|---|---|
| $D_0 = 0.1$ | 63.10 | 0.005 | 36.66 | 0.007 |
| $D_0 = 0.25$ | 62.73 | 0.004 | 36.47 | 0.006 |
| $D_0 = 0.5$ | 62.97 | 0.005 | 36.61 | 0.006 |
| $D_0 = 0.75$ | 63.28 | 0.006 | 36.79 | 0.007 |

### A.3.6 The comparison experiments on sFID and MUSIQ.

As shown in Table 9, we conduct experiments with MUSIQ Ke et al. (2021) and sFID Nash et al. (2021). In image generation at 4x resolution on SDXL, we achieves lower sFid and higher MUSIQ compared to Scale-Crafter, indicating that our method generates actual high-fidelity details.

Table 9: More metrics.

| Method | sFID | MUSIQ |
|---|---|---|
| ScaleCrafter | 1.1025 | 51.65 |
| **Our** | **1.0946** | **52.41** |

### A.3.7 User Study

For the user studies, we utilize SDXL to generate 2048×2048 images, comparing our proposed method against Scale-Crafter (He et al., 2023) and FouriScale (Huang et al., 2024). We randomly select 100 prompts and generate 100 sets of images. Ten volunteers participated in the test, and they were asked to select the image with the best details and a reasonable semantic layout from each set. As shown in Table 10, the results indicate that our method demonstrates significantly superior performance, garnering 50% of the user preference votes. FouriScale and ScaleCrafter followed with 30% and 20% of the votes, respectively.

Table 10: User study.

| Method | Preference |
|---|---|
| ScaleCrafter | 20% |
| FouriScale | 30% |
| **Our** | **50%** |

### A.3.8 More visual results

Our method is plug-and-play for diffusion models. To validate its effectiveness across different model architectures and resolution configurations, we provide additional plug-and-play comparative experimental results, as shown in Fig. 10, 12, 13, 14, 15, 16, 17, 18, 19.

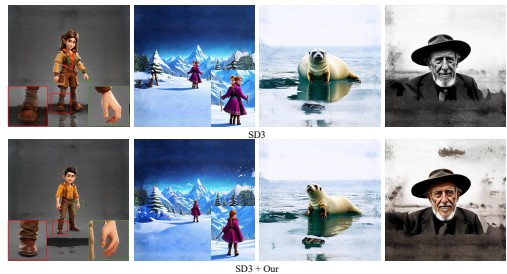

Figure 10: Visual Results on SD3. From left to right, the prompts used in the examples are: (1) "Stylized Character Rendering". (2) "mountain scene from frozen 2". (3) "Greg Lecoeur - Crabeater seal". (4) "old man with a hat by azatyeman".

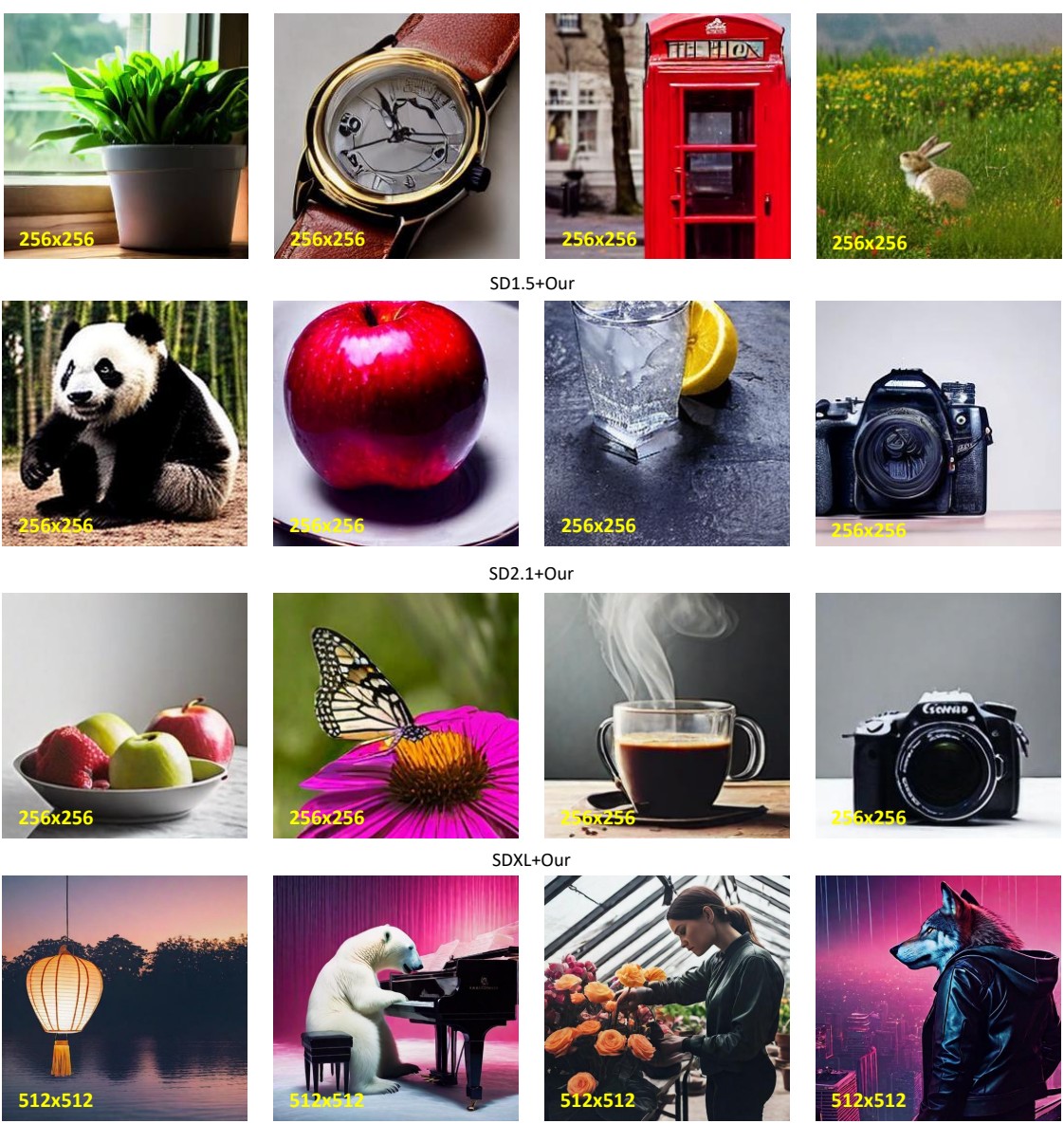

Figure 11: Visual results on lower resolution.

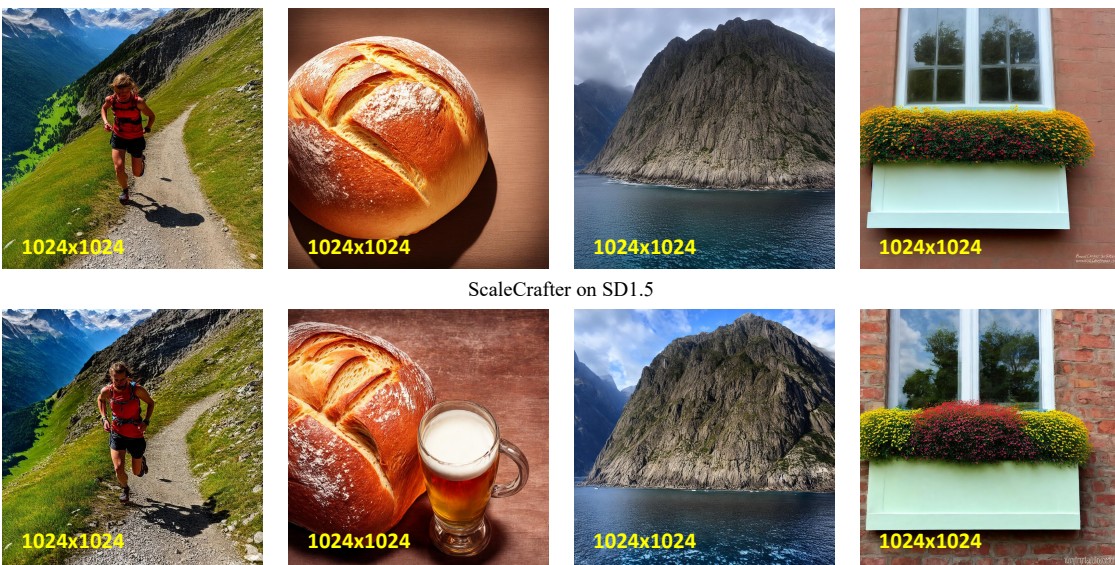

ScaleCrafter on SD1.5

ScaleCrafter+Our on SD1.5

Figure 12: Comparative visualization of generated results on SD1.5 1024×1024 setting. From left to right, the prompts used in the examples are: (1) "Trail running in changing weather while in the Dent du Morcles area of Switzerland". (2) "Bread and Belgian Beer". (3) "Hamnoy Epilogue". (4) "Window Box Painting".

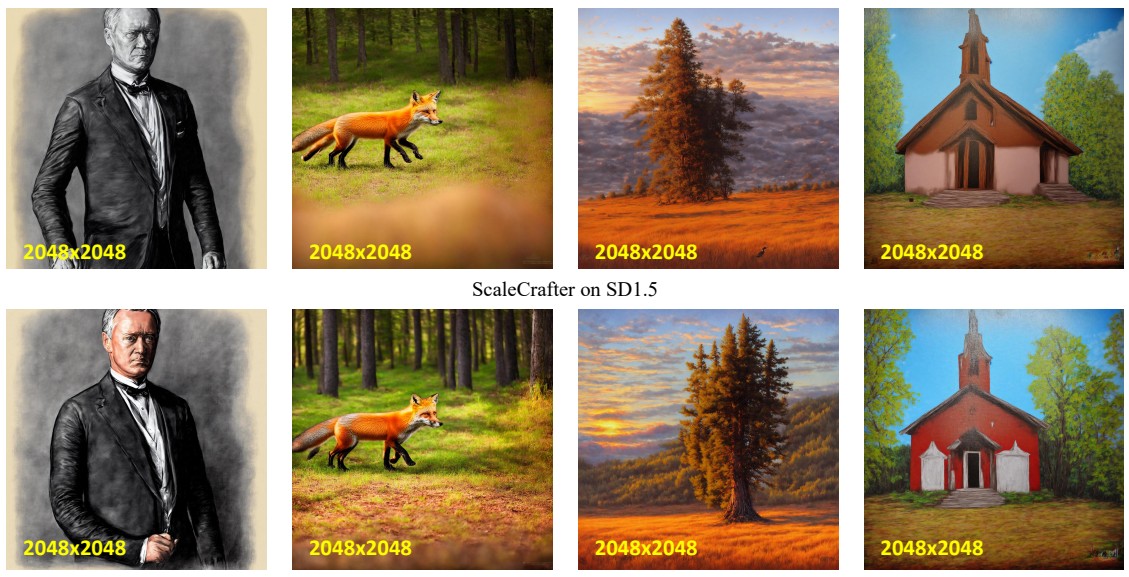

ScaleCrafter on SD1.5

ScaleCrafter+Our on SD1.5

Figure 13: Comparative visualization of generated results on SD2.1 2048×2048 setting. From left to right, the prompts used in the examples are: (1) "Francis J. Underwood by alfalert". (2) "A red fox trotting through a forest, its fur glowing golden in the soft sunlight filtering through the trees.". (3) "Sunrise Sentinel by Martin Grelle". (4) "Chapel Painting - Chapel At Ojo Claiente by Steven Boone".

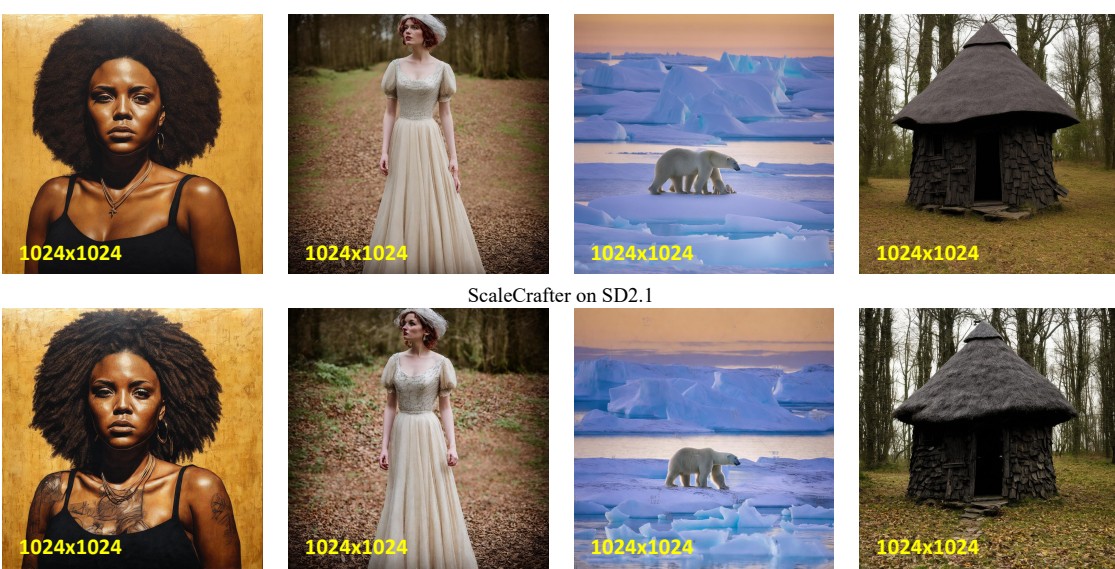

ScaleCrafter on SD2.1

ScaleCrafter+Our on SD2.1

Figure 14: Comparative visualization of generated results on SD2.1 1024×1024 setting. From left to right, the prompts used in the examples are: (1) "Striking Portraits Featuring Powerful Women of Color Painted by Artist Tim Okamura". (2) "Vintage style beaded wedding dress by Joanne Fleming Design, image by Rob Howarth". (3) "Canada, Nunavut Territory, Repulse Bay, Polar Bear and young cub (Ursus maritimus) floating clinging to iceberg near Harbour Islands at sunset". (4) "Witch's Hut".

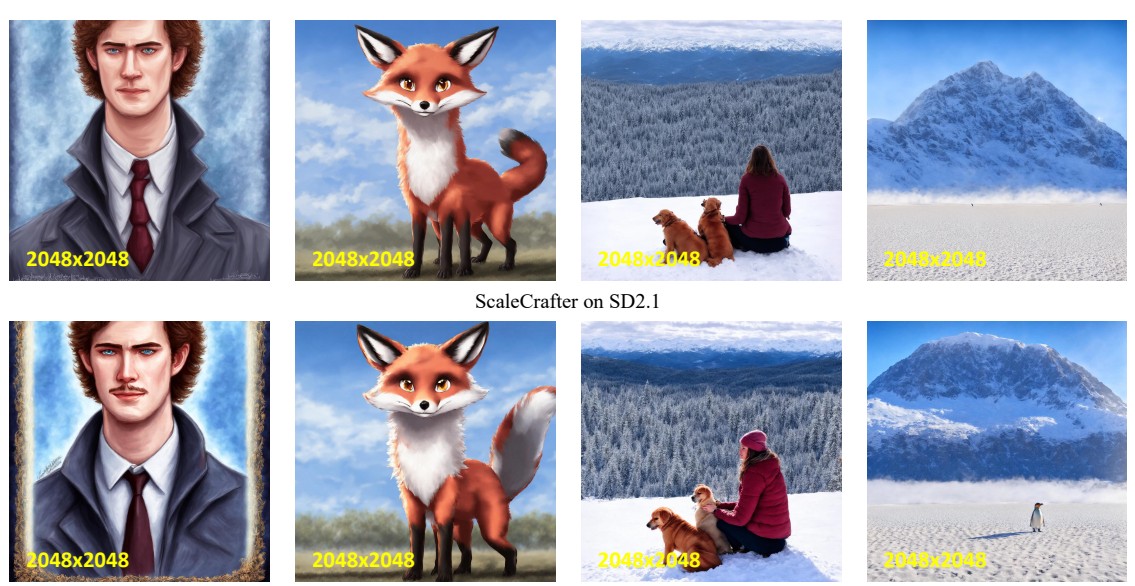

ScaleCrafter on SD2.1

ScaleCrafter+Our on SD2.1

Figure 15: Comparative visualization of generated results on SD2.1 2048×2048 setting. From left to right, the prompts used in the examples are: (1) "Lunar Chronicles - Captain Carswell Thorne by LauraHollingsworth". (2) "e926 2017 aircraft anthro blacknose canine clothed clothing day detailed background digital media (artwork) dipstick tail fox mammal multicolored tail outside sky smile stanidng thanshuhai". (3) "A woman and dog sitting in the snowy mountains". (4) "standing penguin on sand near snow covered mountain covering the sun from view at daytime0".

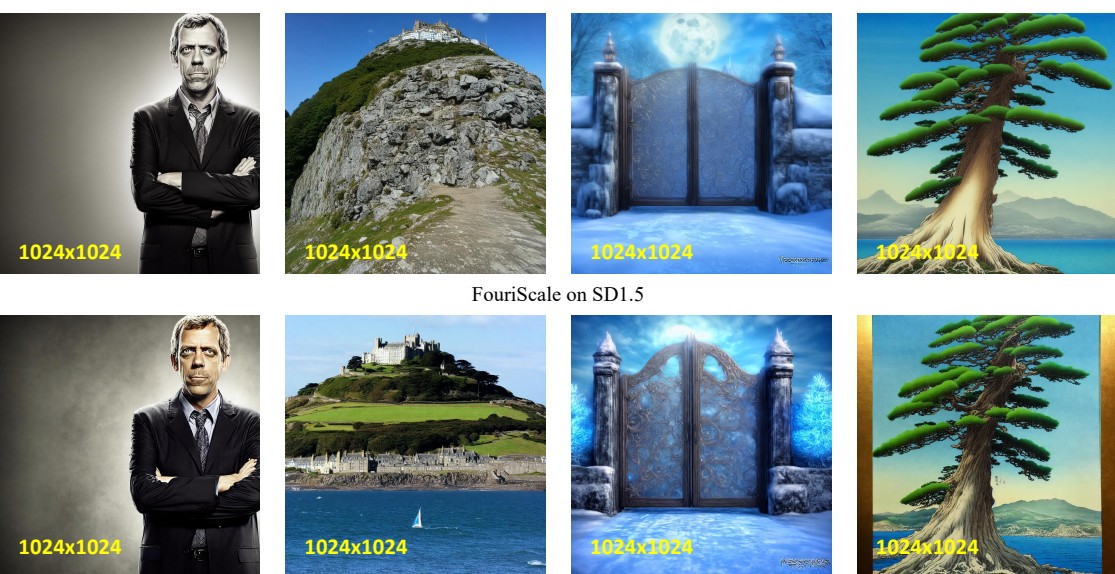

FouriScale+Our on SD1.5

Figure 16: Comparative visualization of generated results on SD1.5 1024×1024 setting. From left to right, the prompts used in the examples are: (1) "Dr. House - Hugh Laurie". (2) "St Michael's Mount". (3) "Frozen gate Tera by moonworker1". (4) """A Lone Carmel Cypress - Original Painting"" by Obata, Chiura".

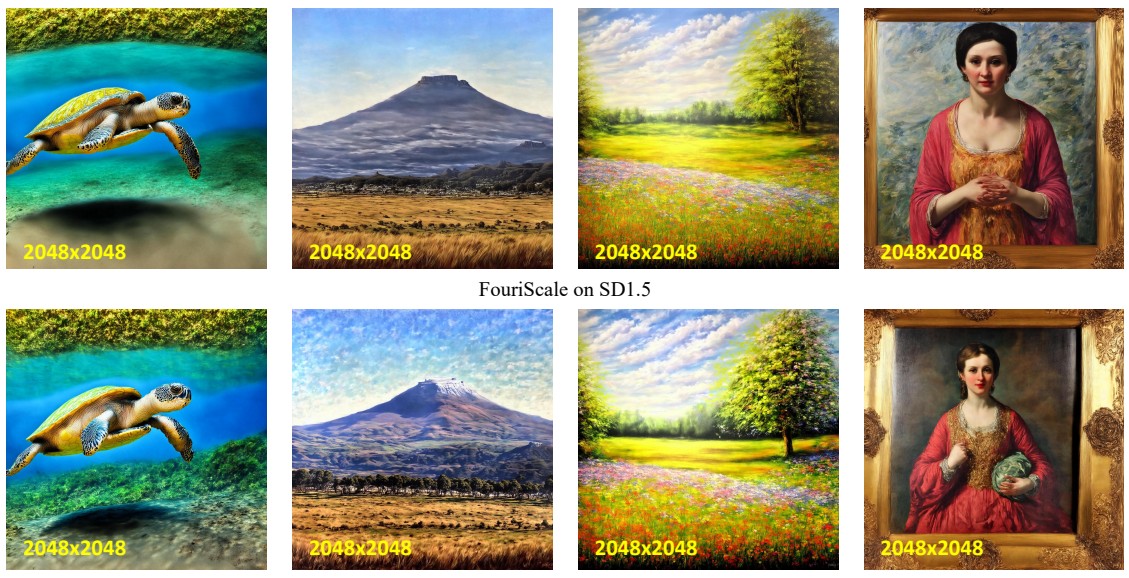

FouriScale+Our on SD1.5

Figure 17: Comparative visualization of generated results on SD2.1 2048×2048 setting. From left to right, the prompts used in the examples are: (1) "Photograph Tropical paradise with turtles by Vitaliy Sokol on 500px". (2) "Plein air oil painting of the rural landscape looking across to Table Mountain from Dysart, Tasmania. By artist Rick Crossland". (3) "Oil Canvas Painting Spring Meadow with Colorful Flowers and Tree". (4) "MacDOUGALL's RUSSIAN ART AUCTION 30 MAY 2020".

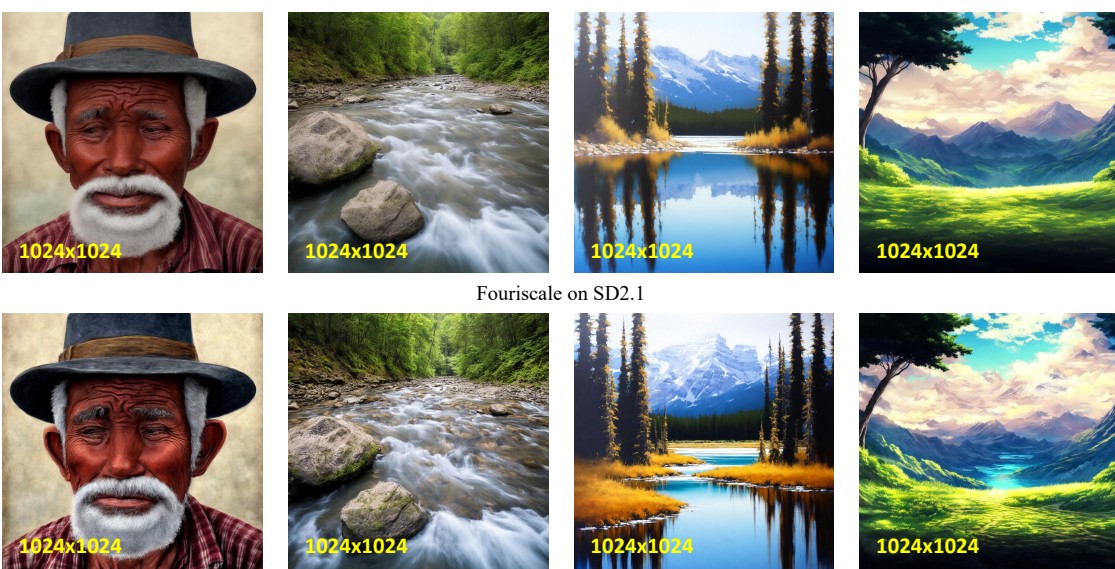

Figure 18: Comparative visualization of generated results on SD2.1 1024×1024 setting. From left to right, the prompts used in the examples are: (1) "old man with a hat by azatyeman". (2) "Tips for Photographing Rivers". (3) "Mike Svob artwork 'STILL WATERS' available at Canada House Gallery - Banff, Alberta". (4) "Anime Landscape 4k Laptop Full Hd 1080p Hd 4k Wallpapers".

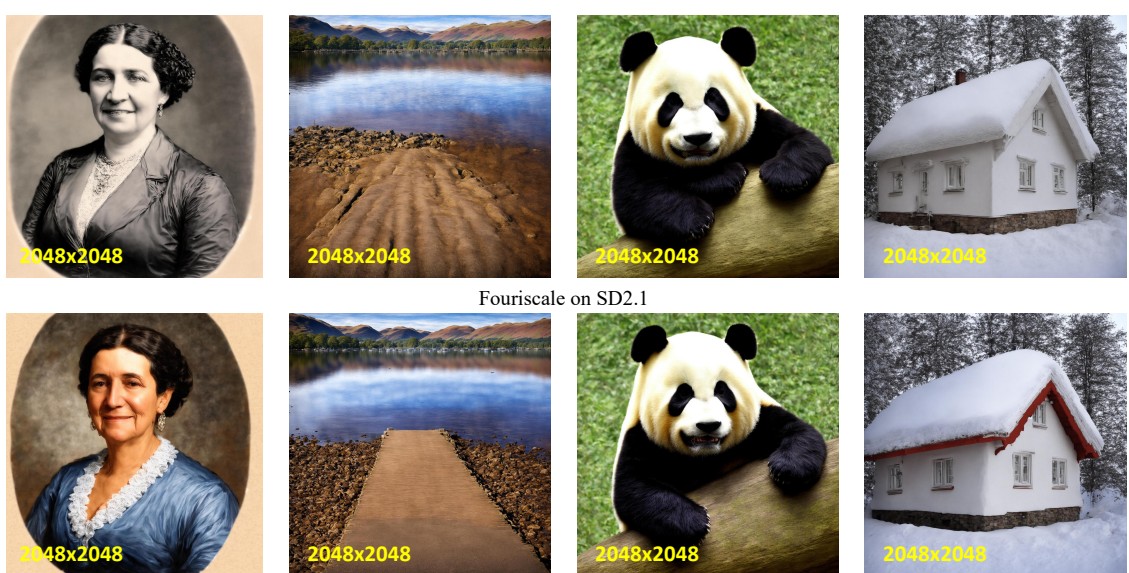

Figure 19: Comparative visualization of generated results on SD2.1 2048×2048 setting. From left to right, the prompts used in the examples are: (1) "Susanna Madora Salter was the first woman elected to political office in the United States. She was elected mayor of Arg". (2) "Stock photo of Ashness Jetty, Derwentwater, Keswick, Lake District National Park". (3) "1789 download wallpaper Animals, Pictures, Pandas screensavers and pictures for free". (4) "Typical Swedish House Cottage Exterior Sweden House Swedish House".

