# OpenReview forum: "InfoScale: Unleashing Training-free Variable-scaled Image Generation via Effective Utilization of Information"
_TMLR — Accepted by TMLR_

### Review · Reviewer_jzkz · 2025-12-18

**Summary Of Contributions:**

Summary

This paper addresses the issue that pre-trained DMs suffer a significant drop in performance when generating images at resolutions different from their training resolution. The authors propose that the core challenge lies in the change in the amount and distribution of information across different resolutions. The paper introduces InfoScale, a training-free framework composed of three key modules:
1. Progressive Frequency Compensation (PFC): Extracts high-frequency components from previous timesteps to compensate for loss due to dilated convolutions.
2. Adaptive Information Aggregation (AIA): Adjusts attention mechanisms to balance local/global information.
3. Noise Adaptation (NA): Redesigns initial noise distribution to prevent repetition or incompleteness.

Strengths:
1. The proposed method is training-free and plug-and-play
2. Detailed problem diagnosis: this paper provides a principled decomposition of why variable-resolution generation fails by tracing specific failure modes to specific architectural components: (1) high-frequency loss to dilated convolutions, (2) structural incoherence to attention mechanisms, and (3) repetition to initial noise entropy—the authors offer valuable theoretical insights that extend beyond just the proposed method.
3. The framework exhibits a clear one-to-one mapping between the identified problems and the proposed solutions.

Weaknesses:
1. The evaluation relies heavily on standard FID and KID. Since these metrics require resizing images to 299x99, they effectively destroy the high-frequency details the authors claim to improve. Furthermore, standard FID is insensitive to the geometric distortions (stretching) and object duplication artifacts that are the primary failure modes in variable-resolution generation.
2. While the visual samples provided are compelling, they represent anecdotal "best-case" scenarios. Without a large-scale human evaluation or robust perceptual metrics computed over a large dataset, it is impossible to determine if these visuals represent the model's average performance or cherry-picked successes.
3. While Appendix A.3.2 includes experiments on SD3 (a Diffusion Transformer), the main paper focuses almost exclusively on U-Net architectures (SD1.5/SDXL). It is theoretically unclear how the PFC module functions within the Transformer-based SD3 architecture, or if it was disabled for those experiments.

**Audience:**

Yes

**Audience Explanation:**

The problem of fixed-resolution constraints in Diffusion Models is a major hurdle for practical deployment. However, the audience requires a rigorous evaluation standard to ensure the proposed solution is robust and generally applicable, not just effective on selected samples.

**Claims And Evidence:**

No

**Claims Explanation:**

While the theoretical analysis is sound, the empirical evidence has significant gaps regarding measurement and statistical reliability:
1. The provided visual samples demonstrate the potential effectiveness of the model, but they do not constitute statistical evidence. The lack of a large-scale human evaluation leaves a gap between the claimed reliability and the demonstrated best-case scenarios.
2. The reliance on Inception-based metrics (FID/KID) means the authors are evaluating downsampled thumbnails rather than actual high-resolution outputs.

**Requested Changes:**

1. The visual samples are insufficient to prove generalizability. The authors should conduct a Human Evaluation Study with a statistically significant sample size (e.g., >100 image pairs, multiple raters).
2. Use more appropriate automatic evaluation metrics
    (1). Report MUSIQ (Multi-Scale Image Quality) scores. This is a reference-free metric that assesses aesthetic quality at native resolution without resizing, crucial for proving that InfoScale generates actual high-fidelity details.
    (2). Report sFID (Spatial FID) for non-square aspect ratios. Unlike global FID, sFID captures spatial features and can better detect if objects are being distorted.

3. Since the field has shifted to DiT (SD3, Flux) and AR, the Appendix results on SD3 are critical to the paper's relevance. Please move the discussion of DiT applicability to the main text. Also suggest to add AR based baselines.

---

### Review · Reviewer_S2oY · 2025-12-27

**Summary Of Contributions:**

The paper proposes InfoScale, a training-free method for variable-scaled image generation (both high-resolution (HR) and low-resolution (LR) ). The authors analyze the degradation in quality of HR and LR generation by using information entropy. Based on this analysis, the authors propose three modules:
1. Progressive Frequency Compensation (PFC) restores high-frequency details lost by dilated convolutions.
2. Adaptive Information Aggregation (AIA) aggregates information in LR generation and balances local and global information for HR generation.
3. Noise Adaptation (NA) redistributes the initial noise using fixed patterns for LR and HR generation.

**Additional Comments:**

### Strengths

-   **[S1] Plug-and-Play Nature:** The method is entirely training-free and integrates into existing pipelines (e.g., "ScaleCrafter + Our" "FreeScale" + Our). Additionally, the authors show results on SD3 in Figure 10 (Appendix).

-   **[S2] Good analysis for Motivation:** The authors present strong analytical results for the motivation of the proposed modules in Section 3.

### Paper-Writing & Presentation:

-   **[P1]** In Figure 6, qualitative example (e.g., "" llama wearing a knitted scarf…"" exhibits poor generations across both the proposed method and baselines. The authors can replace this with a better example which clearly demonstrates the strengths of the proposed method.

-   **[P2] Figure 7 (Ablation Study):** This figure can be improved. Show the full generated images in a single row, followed by a second row with corresponding zoomed-in regions.

**Audience:**

Yes

**Audience Explanation:**

Yes, the findings in this paper will be useful for people working on generative models. Infoscale presents results for U-Net-based DMs, such as SD2.1 and SDXL. Further, they show results on SD3 in the Appendix.

**Broader Impact Concerns:**

As this work builds on current generative methods, the authors should include a broader impact statement and discuss the potential negative societal implications as well.

**Claims And Evidence:**

Yes

**Claims Explanation:**

The authors provide empirical evidence using Information entropy (Equation 1) in Figures 2, 3 and 4. Figure 2 (b) shows the effect of a decrease in average information due to dilated convolution. Similarly, Figure 3 shows that low-resolution generation suffers from insufficient information, and high-resolution generation suffers from redundant information aggregation. Further,  the ablation studies in Table 2 support the proposed modules (PFC, AIA and NA).

**Requested Changes:**

### Weaknesses

-   **[W1] Insufficient discussion on the choice of stop-frequency in PFC:** The authors mentioned that more details can be found in the Appendix (page 7). But the authors do not discuss how sensitive the model is to this value. How does it affect the overall generation process?
-   **[W2] Novelty of Adaptive Information Aggregation (AIA):** In Sec. 4.2, the authors write: "Specifically, following prior work (Jin et al., 2023), we introduce a scaled factor $\alpha$ ...". Attn-SF (Jin et al. 23) already proposes a factor $\lambda$. The only extra empirical hyperparameter is $c_w$ for low-resolution images. However, the authors fix the value of this factor empirically  and its effect is not ablated in the current manuscript.  Further, for HR generation the differentiation with Attn-SF is not clear in the method section. The authors should provide more detail on the fusion of Attn-SF and original attention.
-   **[W3] Assumption in Noise Adaptation Module:** The NA module increases W from the centre to the surrounding in LR generation and decreases W for HR generation. Does this mean that the main semantic content is located in the centre? Will it work for the prompts where a subject is not at the centre, or there are multiple subjects in the scene at different locations? Additional qualitative results with such prompts should be presented.
-   **[W4] Missing User Study:** As a major claim of the paper is superior high-resolution quality, a human user study with sufficient participants will be a better way to evaluate the efficacy of the proposed approach.
-   **[W5] Results on additional resolution scales:** In the current experimental setup, results are only shown for 2x, 3x and 4x for HR generation. What about other factors, such as 8x and 16x? Similarly, for LR generation, only one scale, 0.5x, is shown. What about other scales? Showing results on other scales will strengthen the claims in this paper.


### Questions

-   **[Q1]** How do the proposed techniques compare with the recent Flux-based architectures?
-   **[Q2]** InfoScale proposes PFC, AIA and NA techniques. Can these techniques be applied to conditional generation settings such as inpainting or ControlNet-based models, or are additional modifications required?
-   **[Q3]** Which GPU hardware was used for the evaluation of time performance?
-   **[Q4]** In Fig. 6 (""renaissance-style painting of a cat ..."", the visual difference between ScaleCrafter and ScaleCrafter + Ours is minute and difficult to distinguish. The fur on the cat's neck appears less blurred in the baseline method.

---

### Review · Reviewer_hV7d · 2026-02-10

**Summary Of Contributions:**

The paper proposes a tuning-free approach, InfoScale, that leverages diffusion models trained with fixed resolution to generate images at higher and lower resolutions without fine-tuning.

The paper first identifies three key limitations in previous variable-scale methods: high-frequency information is lost due to dilated convolutions, information aggregation mechanisms, and the initial noise distribution can be improved.
To address these, the paper proposes three modules, Progressive Frequency Compensation (PFC), Adaptive Information Aggregation (AIA), and Noise Adaptation (NA) to tackle the corresponding limitations. The modules can be integrated with existing frameworks like ScaleCrafter and FreeScale.

Experiments on SD1.5, SD2.1, and SDXL demonstrate improvements, but with a very limited margin.
There is no reasonable connection among these three proposed modules. They are simply concatenated and solely address each issue. The paper's workload is increased by adding these three unrelated modules to the approach section. Despite the incremental work, the paper is qualified for the TMLR.

**Audience:**

Yes

**Audience Explanation:**

Researchers can apply this method to generate images at diverse resolutions with better performance on FID and KID.

**Claims And Evidence:**

Yes

**Claims Explanation:**

The paper clearly presents the observed weakness or the need for improvement in this field through information amount analysis.

For every weakness, the method proposes a solution to improve it and conducts experiments to show the improvements on the FID and KID metrics and visualization comparison.

**Requested Changes:**

There are some weaknesses that can be improved.

- Lack of novelty and connection among proposed modules. The three modules are incrementally built upon the existing works [1][2][3] without a clear logic regarding their inter-dependencies. The paper lacks a discussion on how these components interact. For example, in Table 2, the authors need to explain why adding NA to PFC causes the worst KIDc.

- Uniform vs. Gaussian. In Section 3.3, the authors claim "For higher resolution, the spatial distribution of initial noise
is **more uniform (i.e., more like Gaussian)** than the training resolution," and "For lower resolution, the spatial distribution of initial noise is ** less uniform (i.e., less like Gaussian) ** than the training resolution".
This terminology is problematic and confusing. The terms "Uniform" and "Gaussian" are usually different concepts.

- GenEval matric. Noise Adaptation focuses the model on generating content located at the center of the scale. It is better to use GenEval to provide a comprehensive evaluation of the counting and position alignment.

- What is the application and importance of low-resolution generation? If the model works perfectly on high-resolution generation, "High-Res Gen + Downsampling Resize" approach could be a direct way to guarantee correct global structure and high fidelity.

- The caption of Figure 2 is missing.


[1] Fouriscale: A frequency perspective on training-free high-resolution image synthesis

[2] Training-free diffusion model adaptation for variable-sized text-to-image synthesis

[3] Initno: Boosting text-to-image diffusion models via initial noise optimization

---

### Decision · Action_Editor_Yscr · 2026-03-30

**Recommendation:** Accept with minor revision

**Audience:**

Yes

**Audience Explanation:**

This paper addresses the interesting problem of variable-scaled image generation, a popular topic that is currently attracting attention within the generative AI community. Although the specific training-free method proposed may not be broadly applicable beyond future training-free approaches, the insights derived from the method's design and empirical observations could benefit the progressive training of image generation models.

**Claims And Evidence:**

Yes

**Claims Explanation:**

This paper investigates training-free approaches for variable-scaled image generation. It proposes an information-centric method to estimate high-frequency noise, a novel attention mechanism to adjust the aggregation degree of information in self-attention, and a Noise Adaptation module.

Main Claims

* A novel training-free framework for variable-scaled image generation, incorporating three effective modules for this task.

* The effectiveness of the framework as a training-free plug-in for Diffusion Models, verified by empirical experiments.

Evidence for Claims

* Claim 1: All reviewers found the proposed method to be technically sound.

* Claim 2: The empirical results provide solid evidence of the method's efficacy. While one reviewer considered the improvements to be relatively marginal, and another noted that the supportive human evaluation was somewhat small-scale and lacked fine-grained detail, these concerns appear to not undermine the core contributions. The AE views that this claim is backed by sufficient evidence.

Following the rebuttal, there are no remaining major concerns. The only outstanding point raised by reviewers pertains to novelty, which is not a rejection criterion for TMLR submissions.

One reviewer raised a concern regarding the assessment of the method's computational speed. Specifically, the implied speed advantage of InfoScale for low-resolution image generation (suggested in Section 5.1 and Appendix A.1) lacks a rigorous analytical comparison against relevant baselines. Although this is not a main claim of the paper, the AE encourages the authors to clarify this point in the revised manuscript.

---

> ### Comment · Action_Editor_Yscr · 2026-05-08
> **Revision Summary**
>
> Hi Authors, could you please list a summary of changes in the revised manuscript and indicate how each request was addressed? Thanks!
>
> AE